# Complement C3a signaling facilitates skeletal muscle regeneration by regulating monocyte function and trafficking

Congcong Zhang[1], Chunxiao Wang[1], Yulin Li[1], Takashi Miwa[2], Chang Liu[1], Wei Cui[1], Wen-Chao Song[2] & Jie Du[1]

Regeneration of skeletal muscle following injury is accompanied by transient inflammation. Here we show that complement is activated in skeletal muscle injury and plays a key role during regeneration. Genetic ablation of complement C3 or its inactivation with Cobra Venom Factor (CVF) result in impaired muscle regeneration following cardiotoxin-induced injury in mice. The effect of complement in muscle regeneration is mediated by the alternative pathway and C3a receptor (C3aR) signaling, as deletion of *Cfb*, a key alternative pathway component, or *C3aR* leads to impaired regeneration and reduced monocyte/macrophage infiltration. Monocytes from *C3aR*-deficient mice express a reduced level of adhesion molecules, cytokines and genes associated with antigen processing and presentation. Exogenous administration of recombinant CCL5 to *C3aR*-deficient mice rescues the defects in inflammatory cell recruitment and regeneration. These findings reveal an important role of complement C3a in skeletal muscle regeneration, and suggest that manipulating complement system may produce therapeutic benefit in muscle injury and regeneration.

[1] Beijing AnZhen Hospital, Capital Medical University, The Key Laboratory of Remodeling-related Cardiovascular Diseases, Ministry of Education, Beijing Institute of Heart, Lung and Blood Vessel Diseases, Beijing 100029, China. [2] Department of Pharmacology and Institute for Translational Medicine and Therapeutics, University of Pennsylvania, Perelman School of Medicine, Philadelphia, PA 19104, USA. Correspondence and requests for materials should be addressed to W.-C.S. (email: songwe@upenn.edu) or to J.D. (email: jiedu@ccmu.edu.cn)

Skeletal muscle regeneration is an adaptive response to injury or disease (e.g., trauma, Duchenne's muscular atrophy, diabetes etc.) which involves myofiber damage and myogenic satellite cell (SCs) activation, proliferation, differentiation, and fusion to newly formed muscle fibers[1–3]. In addition to autonomous transcriptional regulation, the behavior of SCs is also regulated by the interplay between inflammatory cells and cytokines[4–6]. Recent studies have demonstrated that pro-inflammatory monocytes recruited from peripheral blood functioned as phagocytes and are rapidly converted to anti-inflammatory macrophages that promote the proliferation and differentiation of SCs after injury[7,8]. Depletion of the macrophage population before cardiotoxin injection or after necrotic cell removal all led to impaired muscle regeneration[9]. Several chemokines, such as MCP-1/CCL2, MIP-1α/CCL3, CXCL12, CXCL16, CX3CL1, were reported to participate in muscle injury and regeneration, which could recruit the circulation monocytes to injured muscle and promote the migration of myoblast. However, it is not clear how the increased expression of these chemokines induced in monocyte and muscle cells.

The complement system, which is an important contributor to innate and adaptive immune responses[10], is readily activated during tissue injury to cause inflammation. There are three activation pathways of complement: the classical, the alternative, and the lectin pathways. Complement activation triggers the generation of cleavage products, including the anaphylatoxins C3a and C5a. In addition to their well-documented roles in infection and inflammatory tissue injury, the role of complement proteins in tissue and organ regeneration has been reported in several literatures[11–14]. For example, both C3a and C5a signaling pathways have been found to prime and facilitate liver regeneration after acute carbon tetrachloride injury in mice[15,16]. C3a receptor signaling has been reported to be involved in lens regeneration in the newt[17]. Complement C1q activates canonical Wnt signaling and promotes aging-associated decline in muscle regeneration independent with the classical complement activation[18]. Alternative pathway (AP) of complement activation is required for efficient osteoclast differentiation and regeneration, and complement activation production C3a and C5a could regulate osteoclast differentiation by modulating local IL-6 production[19]. In our previous study using a mouse model of cardiotoxin (CTX)-mediated skeletal muscle injury and regeneration, we obtained microarray data showing increased expression of complement components (C1qa, C3ar1) in injured muscle tissues at 3 days after CTX injection[20]. This suggested a possible role of complement in muscle regeneration but potential mechanisms of such function of complement remain to be delineated.

In this study, we test the hypothesis that complement participates in the regeneration of injured muscle. Here we show that complement is activated early after muscle injury via the AP, and it promotes macrophage-dependent muscle regeneration. By performing bone marrow transplantation and RNA-seq analysis, we further identify complement C3a and C3a receptor signaling as the critical process that mediates macrophage recruitment and muscle regeneration. Exogenous administration of recombinant CCL5 rescues the defects in inflammatory cell recruitment and muscle regeneration of C3aR-deficient mice. These findings reveal an important role of complement C3a pathway in the inflammation initiation and muscle regeneration.

## Results

**Complement activation is critical for muscle regeneration.** We used RNA-seq analysis to profile gene expression in CTX-treated mouse muscle and found that expression levels of a large number of complement genes were changed at 1 day and 2 day after CTX-induced muscle injury (Table 1). Several complement components (Cfb, Cfd, Cfp, C1qa, C1qb, C1qc) had increased expression, whereas expression of some complement regulators (Daf, Cd59a, Serpine2) was decreased (Table 1). Using real-time PCR, we confirmed that C3ar1 and C5ar1, the receptor of complement activation products C3a and C5a, were increased after injury (Fig. 1a). By immune-fluorescence staining, we also detected prominent deposition of activated C3 fragments (C3b/iC3b) in the myofiber of CTX-injured muscle but not in uninjured muscle of naïve mice (Fig. 1b), which provided direct evidence of CTX-

**Table 1 The expression level of complement system genes after muscle injury**

| Gene ID | Symbol | Change fold (Log2) | |
|---|---|---|---|
| | | 1 day/0 day | 2 day/0 day |
| 100609_at | Cfb | 3.91 | 3.88 |
| 93784_at | Cfdp1 | 2.93 | 3.78 |
| 101468_at | Cfp | 3.74 | 4.40 |
| 98562_at | C1qa | 0.17 | 2.54 |
| 96020_at | C1qb | 1.72 | 3.85 |
| 92223_at | C1qc | 0.46 | 2.26 |
| 103707_at | C3ar1 | 3.88 | 5.06 |
| 101728_at | C5ar1 | 2.69 | 2.78 |
| 92198_s_at | Daf2 | −2.25 | −1.09 |
| 101516_at | Cd59a | −2.03 | −2.16 |
| 97487_at | Serpine2 | −1.72 | −2.27 |

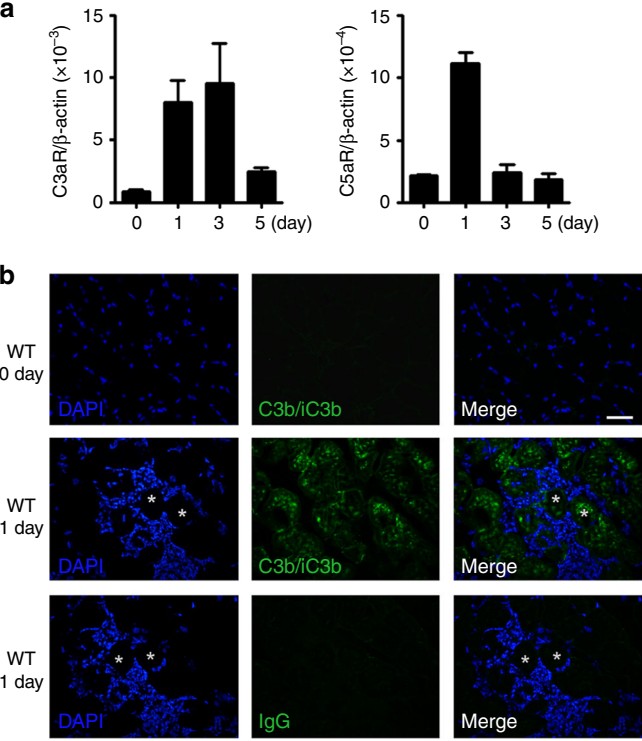

**Fig. 1** Complement system was activated after muscle injury. **a** The mRNA levels of C3aR and C5aR in CTX-injured muscles at day 0, 1, 3, 5 were analyzed by real-time PCR. ($N = 4$ in per time point). **b** Immunofluorescence staining of C3b/iC3b (green) in mouse tissues from normal TA muscle and day 1 after CTX-injured TA muscle, and the nuclei were counter stained with DAPI (blue). IgG antibody was used as the negative control. No C3b/iC3b (green) signal was detected in normal muscle. Deposition of C3b/iC3b (green) was detected in the CTX-injured myofiber. Data shown are representative of three mice in each group

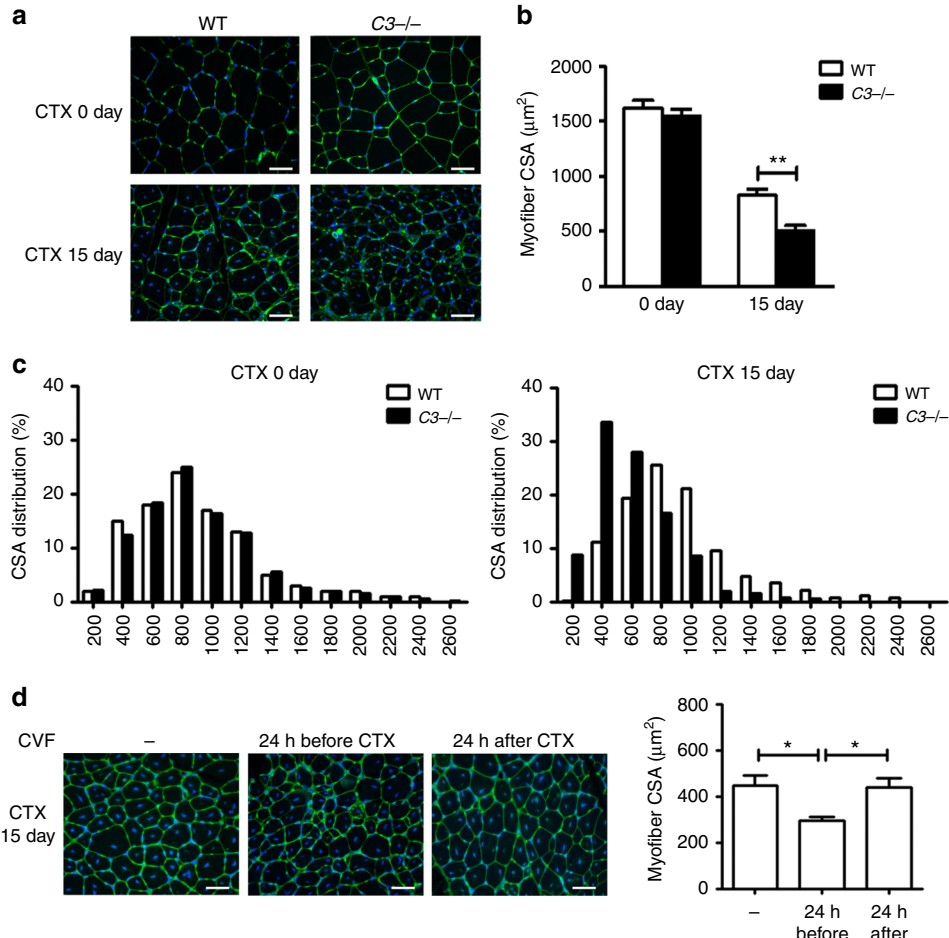

**Fig. 2** Complement activation in the early phase of muscle regeneration. **a** At day 0 and day 15 after injury, muscles from WT and C3−/− mice were immunostained with WGA (green), the nuclei were counter stained with DAPI (blue). (Bars, 50 µm). **b** The mean myofiber CSA in injured muscles from WT and C3−/− mice at 0 and 15 days after injury were measured. (N = 4 in per time point). **c** The distribution of myofiber sizes was analyzed from the CSA of ~250 myofibers of each sample. **d** To deplete serum C3, CVF was injected to WT mice 1 day before or 1 day after CTX injury. At day 15 after injury, muscles from WT mice and CVF-treated mice were immunostained with WGA (green) (bars, 50 µm), the nuclei were counter stained with DAPI (blue). The right graph was the mean myofiber cross section area (CSA) in injured muscles from three groups. (N = 4 in each group). WGA, wheat germ agglutinin; CSA, cross section area. Data are expressed as the mean ± s.e.m. *P < 0.05, **P < 0.01, unpaired t-test, two-tailed

induced local complement activation in muscle tissues. To confirm the role of complement in muscle regeneration after injury, we compared the myofiber cross section area (CSA) of CTX-treated wildtype (WT) and C3−/− mice. Complement C3 is the central protein required for all three pathway of complement activation. As shown in Fig. 2a, b, there was no difference between WT and C3−/− muscle before injury, but the CSA of newly formed myofiber in C3−/− mice was significant smaller than in WT mice at 15 days after CTX injury, and the CSA distribution of C3−/− mouse myofiber was much smaller than WT mice (Fig. 2c). Cobra venom factor (CVF), a functional analog of C3b, forms a stable convertase that depletes serum C3 within hours. Pre-treatment of WT mice with CVF 24 h prior to CTX injection resulted in impaired muscle regeneration as indicated by smaller myofiber CSA of CVF pretreated mice at 15 days after CTX injury (Fig. 2d). However, the treatment did not affect muscle regeneration if given at 24 h after CTX injection (Fig. 2d). These data demonstrated that complement activation played a critical role in the early phase of muscle injury and regeneration.

**The alternative complement pathway is critical for muscle regeneration.** We next determined which complement pathway

(s) is responsible for the impaired muscle regeneration phenotype. Three main pathways can activate the complement system: classical, lectin, and alternative[10]. When antigen-antibody immune complexes binds with C1q and activating C1r and C1s, which further cleave C4 and C2, the classical pathway is activated. When mannose-binding lectin (MBL) activates the MBL-associated serine proteases (MASPs) and further cleaves C4 and C2, the lectin pathway of complement is activated. Cleavage products of C4 and C2 form the classical and lectin pathway C3 convertase (C4bC2a), which cleaves C3 into C3b and C3a. C3b can bind with C4bC2a to form the C5 convertase (C4bC2aC3b). The alternative pathway (AP) of complement is activated when C3 is spontaneously hydrolyzed and forms the initial AP C3 convertase (C3(H2O)Bb) in the presence of Factors B and D, leading to more cleavage of C3 and eventual formation of the AP C3 convertase (C3bBb) and AP C5 convertase (C3bBbC3b). The importance of the classical and lectin pathways was assessed using mice lacking C4. C4−/− mice developed normal muscle regeneration at 30 days after CTX injury, indicating that the classical and lectin pathways were dispensable in this model (Fig. 3a, b). To assess the importance of the AP, we induced muscle injury in mice lacking complement Factor B (Cfb−/−). We found that Cfb deficiency caused a reduction in size of newly formed myofiber at 30 days

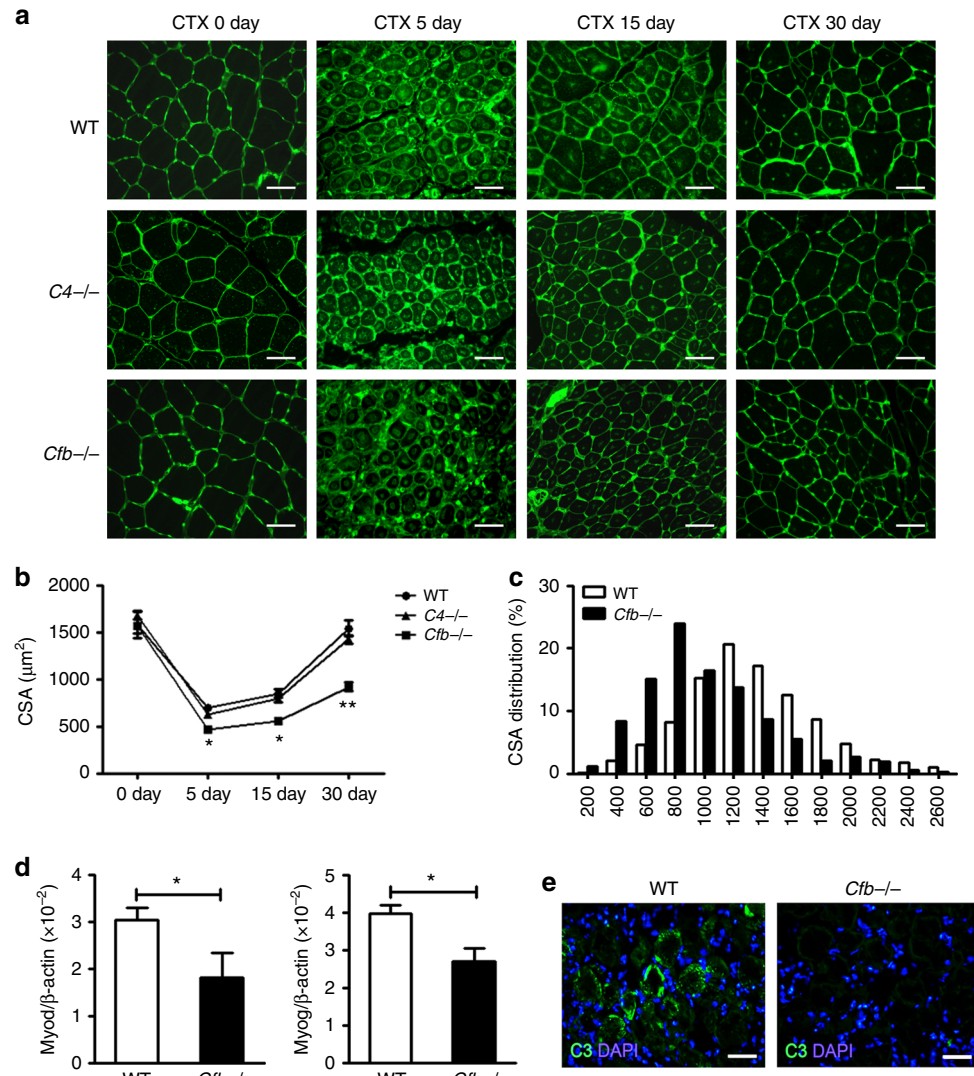

**Fig. 3** The complement alternative activation is critical for muscle regeneration. **a** At 0, 5, 15, and 30 days after injury, muscles from WT, C4−/− and Cfb−/− mice were immunostained with WGA (green). (Bars, 50 μm). **b** The mean myofiber CSA in injured muscles from WT, C4−/− and Cfb−/− mice at 0, 5, 15, and 30 days after injury were measured. (N = 4 in per time point). **c** The distribution of myofiber sizes of WT and Cfb−/− mice at 30 days after injury was analyzed, ~250 myofibers of each sample. **d** At day 5 after injury, the mRNA expression levels of Myod and Myog in WT and Cfb−/− muscle were assessed by real-time PCR (N = 4 in each group). **e** Immunofluorescence staining of C3b/iC3b(green) in WT and Cfb−/− muscle at day 1 after CTX-injured TA muscle, and the nuclei were counter stained with DAPI (blue). (Bars, 50 μm). Data are expressed as the mean ± s.e.m. *P < 0.05, **P < 0.01, unpaired t-test, two-tailed

after CTX injury (Fig. 3a, b), the myofiber CSA distribution was also shifted in Cfb−/− muscle at 30 day after injury (Fig. 3c). Additionally, mRNA levels of Myod and Myog, which are associated with myoblast proliferation and differentiation, were lower in Cfb−/− mice compared with WT mice (Fig. 3d), while mRNA levels of Myod and Myog in C4−/− muscle was similar with WT muscle (Supplementary Fig. 1a). The embryonic MHC mRNA in Cfb−/− muscle, but not in C4−/− muscle, was lower than WT muscle (Supplementary Fig. 1b).

Further supporting the importance of AP, we detected no activated C3 fragment (C3b/iC3b) in injured muscles of Cfb−/− mice, but such products were present in abundant amount in WT mouse injured muscle (Fig. 3e). Based on these findings, we hypothesized that AP complement activation played a beneficial role for efficient muscle regeneration in this model of muscle injury. Thus, we reasoned that restoring Factor B in Cfb−/− mice should lead to AP complement activation and normal muscle regeneration. To test this, Cfb−/− mice were treated with WT

mouse serum as an exogenous source of Factor B. As shown by ELISA assay (Supplementary Fig. 1c), 1 h after intravenous injection of WT mouse serum, LPS-dependent AP complement activity was restored and it lasted 8 h after serum reconstitution. Reconstitution of Cfb−/− mice with WT mouse serum increased Myod and Myog mRNA expression in muscles after CTX injury (Supplementary Fig. 1d) and corrected the defect in muscle regeneration and myoblast proliferation (Supplementary Fig. 1e). In addition, Sirius red staining revealed more interstitial fibrosis formation in CTX-treated Cfb−/− mouse muscle (Supplementary Fig. 2a, b). We also found that mRNA expression levels of Acta2 and Tgfb1, genes that are associated with fibrosis formation, were significantly higher in Cfb−/− mice than in WT mice (Supplementary Fig. 2c), and reconstitution of Cfb−/− mice with WT mouse serum reduced Acta2 and Tgfb1 mRNA levels in the mutant mice (Supplementary Fig. 2d). Taken together, these results suggested that activation of the AP promoted muscle regeneration.

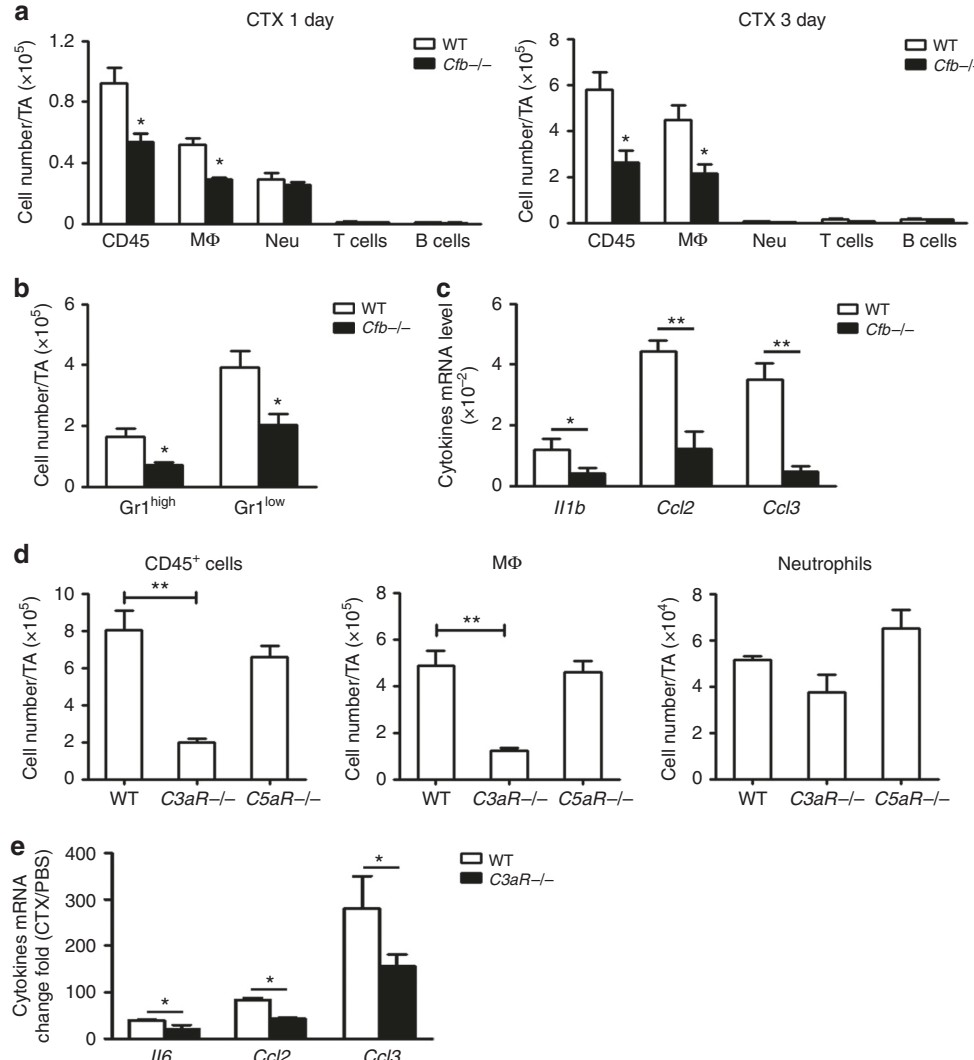

**Fig. 4** Alternative pathway activation and C3aR promote macrophage infiltration. **a** At day 1 and day 3 after injury, the number of CD45+ leukocytes, Gr1+F4/80+ macrophages, Gr1+F4/80− neutrophils, CD3+T cells and B220+B cells in muscles of WT and *Cfb*−/− mice were detected by FACS. The right graph indicated the percentages CD45+ cells of each group (N = 4 in each group). **b** The number of Gr1high and Gr1low macrophages in muscles of WT and *Cfb*−/− mice was also analyzed. (N = 4 in each group). **c** At day 3 after injury, the pro-inflammatory cytokines IL-1β, CCL2, CCL3 mRNA levels in WT and *Cfb*−/− muscles were accessed by real-time PCR. (N = 4 in each group). **d** At day 3 after injury, the number of CD45+ leukocytes, Gr1+F4/80+ macrophages, Gr1+F4/80− neutrophils in muscles of WT, *C3aR*−/−, and *C5aR*−/− mice were detected by FACS. (N = 4 in each group). **e** At day 3 after injury, pro-inflammatory cytokines IL-6, CCL2, CCL3 mRNA levels (changed fold compared with uninjured muscle) in WT and *C3aR*−/− muscles were accessed by qRT-PCR. (N = 4 in each group). Data are expressed as the mean ± s.e.m. *P < 0.05, **P < 0.01, unpaired *t*-test, two-tailed

**AP activation leads to macrophage recruitment after injury**. We next sought to determine the mechanism by which the AP complement contributed to muscle regeneration. We observed that the myoblast proliferation rate, as determined by BrdU incorporation, was lower in *Cfb*−/− mouse muscle than in WT mouse muscle (Supplementary Fig. 1f). Previous studies showed that pro-inflammatory macrophages, infiltrated into muscle tissue immediately after CTX injury, promoted myoblast proliferation[7]. We hypothesized that inactivation of AP complement activation may have impaired macrophage infiltration and muscle regeneration. By using the fluorescence activated cell sorting (FACS) gating strategies (Supplementary Fig. 2e), we found that there were significantly less infiltrating CD45+ leukocytes in *Cfb*−/− mouse muscle than in WT muscle at day 1 and day 3 after CTX injury (Fig. 4a). Furthermore, the number of CD11b+F4/80+ macrophage in *Cfb*−/− mouse muscle was also lower than that in WT muscle, while there was no difference in the number of Gr1

+F4/80− neutrophils, CD3+ T cells and B220+ B cells between WT and *Cfb*−/− muscle (Fig. 4a). Both number of F4/80+Gr1hi pro-inflammatory macrophages and F4/80+Gr1low anti-inflammatory macrophages[8] was also decreased in *Cfb*−/− muscle (Fig. 4b). Indeed, we observed that mRNA levels for several cytokines (*Il1b*, *Ccl2*, and *Ccl3*) were drastically reduced in *Cfb*−/− mouse muscle compared with WT mouse muscle (Fig. 4c). These data suggested that AP complement activation is important for macrophage recruitment which is known to play a facilitating role in the regeneration of injured muscle.

**Complement C3aR deficiency impairs muscle regeneration**. C3a and C5a, the complement activation products known as anaphylatoxins, play a key role in inflammation through C3a receptor and C5a receptor expressed on inflammatory cells. To determine if C3aR or C5aR signaling was involved in macrophage

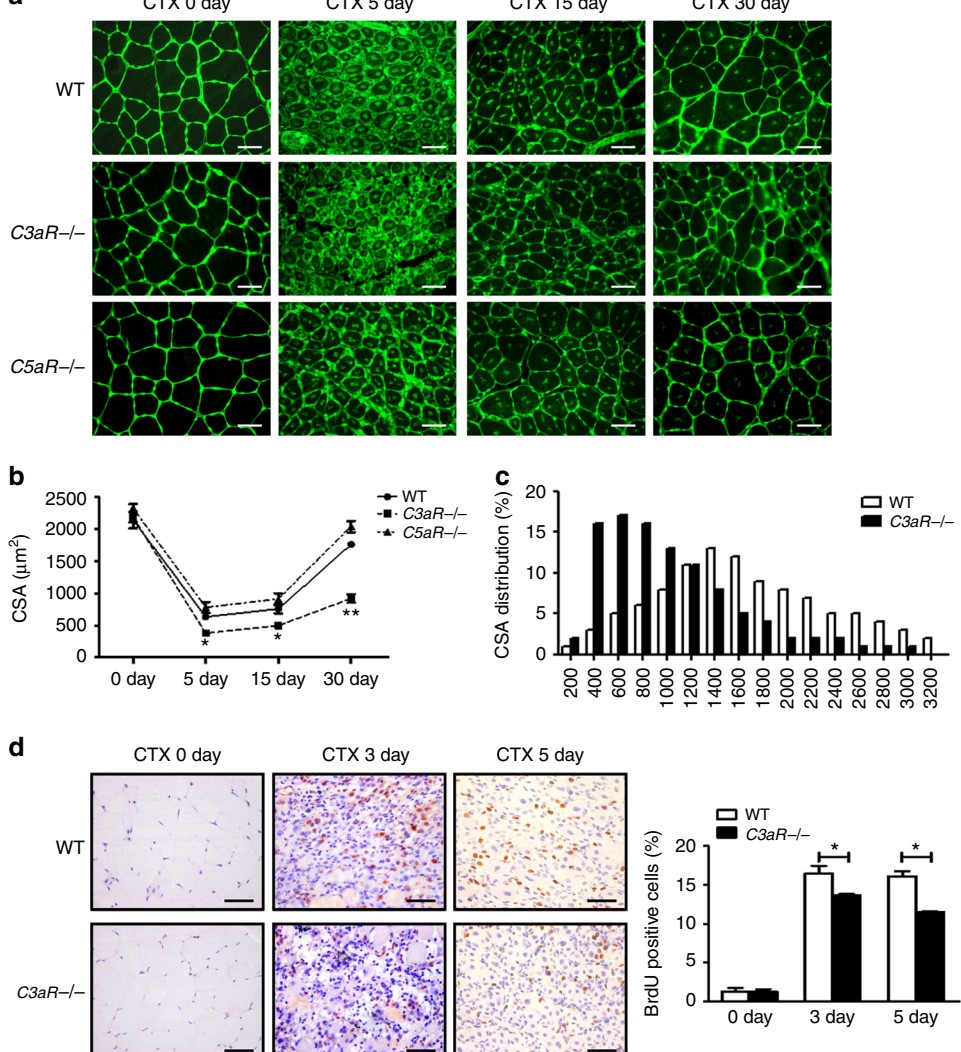

**Fig. 5** Complement receptor C3aR deficiency impaired muscle regeneration. **a** At 0, 5, 15 and 30 days after injury, muscles from WT, *C3aR−/−*, and *C5aR−/−* mice were immune-stained with WGA (green). (Bars, 50 μm). **b** The mean myofiber CSA in injured muscles from WT, *C3aR−/−*, and *C5aR−/−* mice were measured. ($N = 4$ in each group). **c** The distribution of myofiber sizes of WT and *C3aR−/−* mice at 30 days after injury was analyzed, ~250 myofibers of each sample. **d** BrdU immunohistochemical staining was used to detect the proliferating cells in injured muscle (at day 3 and day 5 after injury) of WT and *C3aR−/−* mice, the percentages of BrdU positive cell per field were analyzed. ($N = 4$ in each group; bars, 50 μm). Data are expressed as the mean ± s.e.m. *$P < 0.05$, **$P < 0.01$, unpaired $t$-test, two-tailed

infiltration into injured muscle tissue, we used *C3aR* knockout (*C3aR−/−*) and *C5aR* knockout (*C5aR−/−*) mice in the CTX muscle injury and regeneration model. As shown in Fig. 4d, there was less CD45[+] leukocyte and CD11b[+]F4/80[+] macrophage infiltration in *C3aR−/−* mouse muscle compared with WT mouse muscle at day 1 after injury, but there was no difference in inflammatory cell infiltrates between WT and *C5aR−/−* mouse muscles. The mRNA levels of the pro-inflammatory cytokines (*Il6*, *Ccl2*, and *Ccl3*) were also significantly lower in *C3aR−/−* mouse muscle than in WT muscle (Fig. 4e). Although there was less number of apoptosis cells in *C3aR−/−* mouse muscle than in WT muscle after injury (Supplementary Fig. 3a), flow cytometry analysis revealed no difference in the percentage of apoptotic F4/80[+] Annexin V[+]PI[+] macrophages in the muscle between WT and *C3aR−/−* mice (Supplementary Fig. 3b). In addition, the mRNA levels of genes associated with cell apoptosis (*Bax, Bid, Bcl2, Bcl2l1*) in CD11b[+] cells recovered by FACS sorting from CTX-injured muscle at 3 days after CTX treatment were also similar between WT and *C3aR−/−* mice (Supplementary Fig. 3c). There

was also no difference in the ratio of S phase macrophages and of G2/M phase macrophages from WT and *C3aR−/−* muscle at day 3 after injury (Supplementary Fig. 3d). These data indicated that complement C3a/C3aR signaling affected the recruitment of macrophages rather than the rate of apoptosis and proliferation of such cells in the muscle.

Similar to findings with *Cfb−/−* mice, we found that C3aR deficiency resulted in smaller newly formed myofiber at 15 days and 30 days after CTX injury, whereas C5aR deficiency did not affect the cross section area of newly formed myofiber (Fig. 5a–c). Additionally, Masson staining analysis revealed more interstitial fibrosis formation in *C3aR−/−* mouse muscle sections (Supplementary Fig. 4a). The latter result was supported by data on mRNA levels of *Acta2* and *Tgfb1*, which are genes associated with fibrosis formation. *C3aR−/−* mouse muscle contained higher levels of *Acta2* and *Tgfb1* mRNAs than WT mice (Supplementary Fig. 4b). Myoblast proliferation as measured by BrdU incorporation was also lower in *C3aR−/−* mouse muscle than that in WT mouse muscle (Fig. 5d). In vitro, recombinant C3a did not affect

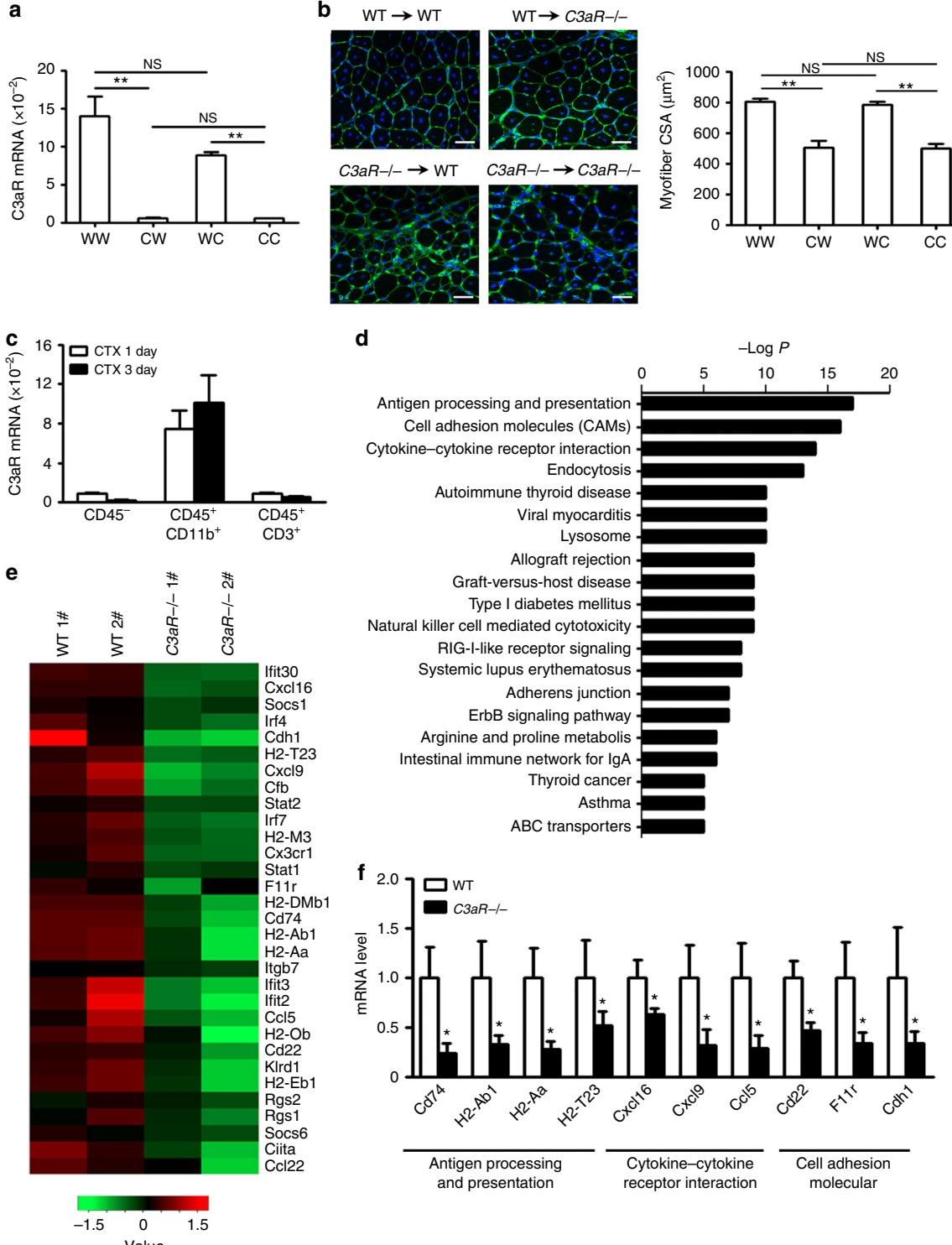

**Fig. 6** Macrophage expressed C3aR mediated macrophage activation and muscle regeneration. **a** C3aR mRNA levels of bone marrow cells from four groups of chimeric mice (WW, WT to WT; CW, *C3aR−/−* to WT; WC, WT to *C3aR−/−*; and CC, *C3aR−/−* to *C3aR−/−*) were accessed by real-time PCR. (*N* = 4 in each group). **b** Four groups of chimeric mice were injured by CTX injection. After 15 days, the regenerating myofiber CSA were examined by WGA staining. The right graph indicated the mean CSA of four group (*N* = 4 in each group, graphs show representatives of two independent experiments). **c** Muscle resident cells (CD45−), monocytes (CD45+CD11b+), and T cells (CD45+CD3+) were isolated from injured muscles (1 day and 3 day after injury) by flow sorting. Then, real-time PCR was used to detect *C3aR* mRNA expression levels of each type of cells (*N* = 3 in each group). **d** CD11b+ cells were sorted from WT and *C3aR−/−* muscle at day 1 after injury, then the transcription expression profiling were accessed by RNA-seq. The KEGG pathway of downregulated genes in *C3aR−/−* cells was analyzed. **e** Some different expressed genes in WT and *C3aR−/−* CD11b+ cells associated with the top three pathways were shown as heat map. **f** Real-time PCR verified the mRNA levels of ten genes associated with the top three changed pathways in WT and *C3aR−/−* CD11b+ cells (*N* = 3 in each group). Data are expressed as the mean ± s.e.m. *$P < 0.05$, **$P < 0.01$, unpaired *t*-test, two-tailed

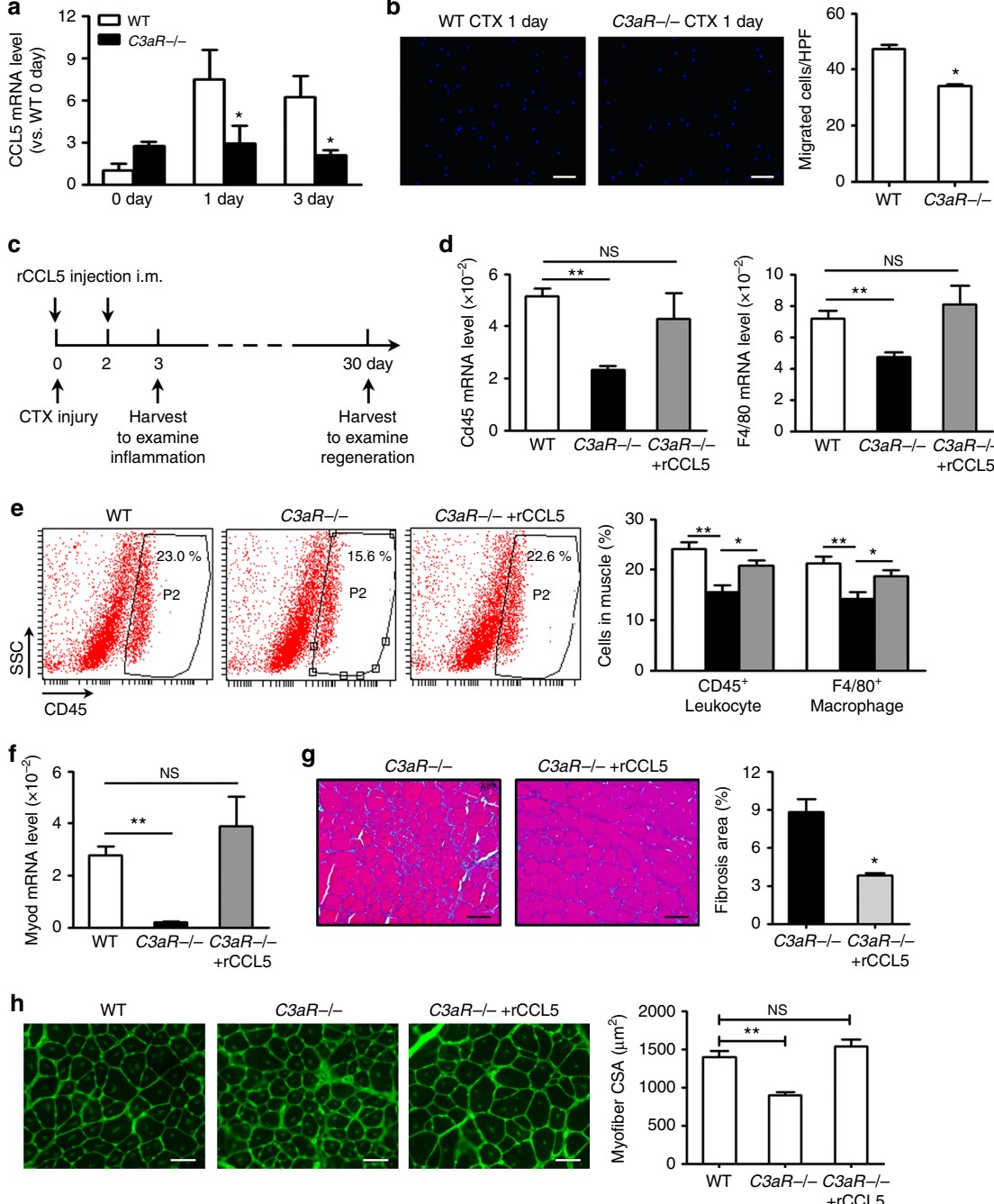

**Fig. 7** Recombinant CCL5 rescued the impaired muscle regeneration of *C3aR−/−* mouse. **a** At day 0, 1, 3 after injury, mRNA levels of cytokines CCL5 in WT and *C3aR−/−* muscles were accessed by real-time PCR. (*N* = 4 in each group). **b** At day 1 after CTX injury, WT (*N* = 8) and *C3aR−/−* (*N* = 4) muscle extract were seeded in the lower chamber and WT monocytes (5 × 10⁴/well) were seeded in the upper chamber. Twelve hours after co-culture, migrated cells from upper chamber were counted by DAPI staining (blue). (Bars, 50 μm, ten filed was collected per sample). **c** Schematic diagram of the protocol. Recombinant mouse CCL5 (0.1 mg kg⁻¹) was intramuscular injected to *C3aR−/−* muscle at day 0 and day 2 after CTX injury. At day 3 after injury, TA muscle was harvested to examine the inflammation response, and at day 30 after injury, TA muscle was harvested to examine the regeneration. **d** mRNA levels of CD45 and F4/80 in WT, *C3aR−/−* and *C3aR−/−* with rCCL5 muscles were accessed by real-time PCR. (*N* = 4 in each group). **e** CD45⁺ leukocytes and F4/80⁺ macrophages in muscles were detected by FACS. The right graph indicated the percentages of CD45⁺ leukocytes and F4/80⁺ macrophages (*N* = 4 in each group, graphs show representatives of two independent experiments). **f** mRNA levels of MyoD in WT, *C3aR−/−* and *C3aR−/−*with rCCL5 muscles were accessed by real-time PCR. (*N* = 4 in each group). **g** Masson staining was used to detect the collagen deposition (green color) in muscle (at day 15 after injury) of WT and *C3aR−/−* mice, the percentages of fibrosis area per field were analyzed. (Bars, 50 μm; *N* = 4 in each group). **h** The mean myofiber CSA in injured muscles from WT, *C3aR−/−* and *C3aR−/−* with recombinant CCL5 mice at day 30 after injury were measured. (*N* = 4 in each group, bars, 50 μm). Data are expressed as the mean ± s.e.m. *\*P* < 0.05, *\*\*P* < 0.01, unpaired *t*-test, two-tailed

the proliferation of myoblast (Supplementary Fig. 5a). To further confirm the role of C5a–C5aR signaling in muscle regeneration, we used an anti-C5 monoclonal antibody to inhibit the generation of C5a during muscle injury and regeneration. At 15 days after CTX injury, we found that muscle regeneration was not affected by the treatment with the anti-C5 monoclonal antibody (Supplementary Fig. 5b). We also performed a glycerol induced muscle injury in WT and *C3aR−/−* muscle, and examined the muscle regeneration and fibrosis at 15 day after injury. As the data shown below, the CSA of regenerated myofiber was much smaller in *C3aR−/−* muscle than that of WT muscle, and the fibrosis area increased in *C3aR−/−* muscle (Supplementary Fig. 5c–e). Collectively, these results suggested that complement C3a–C3aR pathway but not C5a–C5aR pathway promoted macrophage recruitment and muscle regeneration.

**Macrophage expressed C3aR contributes to muscle regeneration.** To determine the cell type on which C3aR is expressed and functional in the injured muscle, we generated bone marrow chimera mice between WT and *C3aR−/−* mice. To confirm the efficiency of bone marrow reconstitution, *C3aR* mRNA level was accessed in bone marrow cells of four types bone marrow chimera mouse, the *C3aR* mRNA was higher in WT and *C3aR−/−* recipient mice with WT bone marrow than that with *C3aR−/−* bone marrow (Fig. 6a). By treating different chimers with CTX and studying for muscle injury and regeneration, we confirmed that C3aR deficiency on bone marrow cells was sufficient to recapitulate the muscle regeneration defect observed in *C3aR−/−* mice (Fig. 6b). Then we sorted out muscle resident CD45⁻ cells (which include CD31⁺ endothelia cells, CD31⁻Sca-1⁺ fibro/adipogenic progenitors, CD31⁻α7-integrin⁺ myoblast), CD45⁺CD11b⁺ monocytes, and CD45⁺CD3⁺ T cells from CTX-injured WT muscles at day 1 and day 3 after CTX injury and examined the mRNA levels of *C3aR* by real-time PCR. We found that both at day 1 and day 3, the CD45⁺CD11b⁺ monocytes population expressed the highest *C3aR* mRNA level among these cells (Fig. 6c).

To investigate the mechanism by which complement C3a–C3aR signaling contributed to macrophages infiltration, leading to a stimulatory effect on muscle regeneration, anti-CD11b magnetic beads were used to sort infiltrated macrophages and neutrophils from WT and *C3aR−/−* mouse muscles at day 1 after injury, and profiled mRNA expression by RNA-seq and bioinformatics approach. There were 154 genes that were altered significantly (>1.5-fold change) between WT and *C3aR−/−* cells. Among them, 67 genes were upregulated and 87 genes were downregulated. Based on Kyoto Encyclopedia of Genes and Genomes (KEGG) pathway analysis, the top three groups were genes involved in antigen processing and presentation, cytokine–cytokine receptor interaction and cell adhesion (Fig. 6d). A heat map showing marked downregulation of genes related to the top three altered pathways in *C3aR−/−* macrophages is depicted in Fig. 6e. By real-time PCR, we analyzed 15 downregulated genes from the list and were able to confirm ten of them with significantly lower expression in *C3aR−/−* monocytes/macrophages (Fig. 6f). These data indicated that monocytes/macrophages expressed C3aR which mediated the activation and amplification of inflammation in injured muscles.

**Recombinant CCL5 rescues impaired regeneration of *C3aR−/−* mice.** Gene interaction network analysis showed that among the downregulated genes, *Ccl5* is directly connected *C3aR* and other genes (Supplementary Fig. 6). We found that in vitro, recombination C3a and C3aR agonist stimulation could increase the mRNA and protein levels of CCL5 in bone marrow-derived

monocytes (Supplementary Fig. 7a, b). Moreover, C3aR activation increased the phosphorylation of AKT and NF-κB, which were associated with CCL5 transcription (Supplementary Figs 7c and 8). We found the *Ccl5* mRNA level was lower in *C3aR−/−* mouse muscle than in WT mouse muscle at day 1 and day 3 after injury (Fig. 7a). To add the evidence of the link between C3a/C3aR and CCL5 in the process of monocyte recruitment, transwell co-culture experiment was performed. C3aR-activated WT monocytes could recruit more monocyte from upper chamber than *C3aR−/−* monocytes (Supplementary Fig. 7d). After CCL5 was neutralized by anti-CCL5 antibody (10 μg ml⁻¹), the C3aR-activated WT monocytes recruited less monocytes (Supplementary Fig. 7e). The muscle extraction from *C3aR−/−* mouse recruited less monocytes than WT muscle (Fig. 7b), the CCL5 concentration of *C3aR−/−* muscle extract culture supernatant was also less than that of WT muscle (Supplementary Fig. 7f). To further examine the role of CCL5 in the muscle regeneration defect of *C3aR−/−* mice, we injected recombinant mouse CCL5 (0.1 mg kg⁻¹) into *C3aR−/−* mouse muscle at day 0 and day 2 after CTX injury (Fig. 7c). Exogenous CCL5 administration increased CD45 and F4/80 mRNA levels in *C3aR−/−* muscle (Fig. 7d). The data was corroborated by the increased infiltration of CD45⁺ leukocytes and F4/80⁺ macrophages into *C3aR−/−* muscle (Fig. 7e). Most importantly, the defects observed in *C3aR−/−* mouse muscle in muscle regeneration as measured by mean myofiber cross-sectional area, *Myod* mRNA expression and muscle fibrosis were rescued by CCL5 administration (Fig. 7f–h). Thus, complement C3a–C3aR signaling stimulated the expression of cytokines and chemokines, which then contributed to macrophages recruitment after muscle injury to facilitate muscle regeneration.

## Discussion

In this study, we identified complement as a critical mediator of macrophage recruitment and subsequent muscle regeneration after injury. We made several new findings: (1) the activation of the complement cascade promotes muscle regeneration after injury; (2) the AP plays a major role in the process while the classical and lectin pathways were apparently not critical; and (3) complement C3a–C3aR, but not C5a–C5aR, signaling is required for initiating recruitment of circulating monocytes into injured muscle where they promoted muscle regeneration.

Several lines of evidence supports our conclusion that complement plays a key role in regeneration of CTX-injured muscle. We found abundant deposition of C3b in injured muscle tissue, indicating that the injured muscle activated the complement system. When C3, the central component of the complement system, was genetically deficient or exhausted by CVF treatment in mice, CTX-injured muscle had impaired regeneration as indicated by smaller newly formed myofibers in such animals, confirming that activated complement could promote muscle regeneration after injury. Interestingly, when C3 depletion by CVF was performed at day 2 after CTX muscle injury, no effect on muscle regeneration was observed, suggesting that complement activation during the early phase of muscle injury was critical for its beneficial role. Immediately after CTX treatment, damaged muscle fibers could release a number of damage-associated molecular patterns, including intracellular proteins, ATP[21], heat shock protein (HSP)[22] or high mobility group box-1 protein (HMGB1)[23], as well as non-protein components derived from the extracellular matrix such as hyaluronan fragments[24], all of which were potential activators of complement. The deposition of complement activation fragment C3b on damaged myofiber was consistent with the hypothesis that the complement cascade was activated by injured myofiber.

A natural question is the pathway by which complement is activated in CTX-injured muscle. Complement activation can proceed via three pathway, classical pathway, lectin and AP. Our study found that blocking classical and lectin pathways by *C4* gene knockout did not have any effect on muscle regeneration, whereas blocking the AP by *Cfb* gene knockout resulted in disappearance of C3b deposition on damaged myofiber and impaired muscle regeneration. Reconstitution of *Cfb*−/− mice with WT mouse serum rescued the phenotype, confirming the critical role of the AP in muscle regeneration. This finding is consistent with other published studies demonstrating that HSP and HMGB1 released from damaged cells could induce complement activation through the AP[22,23]. In another published study, it was shown that neo-antigen expressed on damaged tissues could be recognized by IgG, which likewise activated the AP of complement[25]. Our results are consistent with these previous observations, but the exact nature of molecules that activated the AP complement in CTX-injured muscle remains to be determined.

In addition to being negatively affected by *Cfb* deficiency and AP complement activity, muscle regeneration after CTX injury was also impaired by *C3aR* deficiency, suggesting a role of C3a–C3aR signaling. We found that C3a had no direct effect on myoblast proliferation in vitro, suggesting that the promoting effect of activated complement on muscle regeneration was not directly on myoblasts. Instead, we established that activated complement acted by promoting macrophage infiltration into injured muscle tissue. Several chemokines, such as MCP-1/CCL2[26,27], MIP-1α/CCL3[28], CXCL12[29], CXCL16[30], CX3CL1[31], was reported to participate in muscle injury and regeneration, which could recruit the circulation monocytes to injured muscle or promote the migration and proliferation of myoblast. But how the increased expression of these chemokines induced in monocyte and muscle cells was still unknown. Our previous study found that cytokine interleukine-6 could increase the expression of CCL2 and CCL3 in macrophages, which could promote muscle regeneration[32]. In this study we have found that complement system was activated immediately after muscle injury, and complement C3a could increase the expression CCL5 in monocytes, which could recruit more macrophages to injured muscle. The decreased chemokines expression could resulted in decreased infiltration of macrophages, which was often accompanied by impaired myoblast proliferation and muscle regeneration[26,33]. In the present study, we found the number of infiltrating macrophages in *Cfb*−/− and *C3aR*−/− muscle to be less than in WT muscle, and several inflammation- and cytokine-associated genes were differentially expressed in WT and *C3aR*−/− monocytes. These results all suggested that the role of activated complement in muscle regeneration was exerted through inflammation regulation. Thus, among the three categories of complement activity by activated products, namely anaphylatoxin C3a and C5a regulated inflammatory response, opsonization with C3b to facilitate phagocytosis, and membrane attack complex (C5b6-9) mediated cell lysis, C3a–C3aR signaling on macrophages to stimulate its activation and trafficking into the muscle appeared to be the dominant mode of action. Indeed, we found the number of infiltrating macrophages in *Cfb*−/− muscle was decreased, but there was no difference in the clearance of damaged myofiber, indicating again the main role of complement in this model was to promote the infiltration of macrophages.

It is interesting that we observed a clear role for C3a–C3aR signaling but not C5a–C5aR signaling in muscle regeneration. Lack of an effect by C5a–C5aR signaling was also confirmed by the experiment using an anti-C5 monoclonal antibody. Bioinformatics analysis of RNA-seq experiment showed that genes associated with antigen processing and presentation, cell adhesion and cytokine–cytokine receptor interaction were significantly downregulated in *C3aR*-deficient mouse monocytes/macrophages. This finding indicated that C3a–C3aR signaling was important for initiation and amplification of inflammation after muscle injury. We identified one of the key downstream target genes to be *Ccl5*, which occupies a central position in a constructed C3aR-related gene network. CCL5 is capable of recruiting immune cells, particularly T lymphocytes and monocytes/macrophages[34], in many settings and our experiment also demonstrated that exogenous CCL5 administration rescued the defect in muscle regeneration of C3aR-deficient mice through stimulation of leukocyte recruitment.

Furthermore, the cellular/molecular mechanisms that can potentially link complement, macrophages, and regeneration are explored in this study. C3a receptor C3aR is a G protein coupled receptor. C3aR activation could increase the phosphorylation of AKT and NF-κB as downstream signaling pathway, which was reported to promote the transcription of chemokine CCL5[35]. Chemokine CCL5 could recruit the macrophages into injured muscle, the decreased CCL5 in *C3aR*−/− muscle recruited less macrophages, which was evidenced by the transwell co-culture experiment. The infiltrated macrophages could secrete other chemokines to recruited more macrophages, which amplify the inflammatory response in injured muscle. The infiltrated macrophages was reported to facility muscle regeneration by clearing the injured muscle by phagocytosis and producing growth factors and cytokines to promote myoblast proliferation and differentiation[7,36].

In conclusion, this study provides evidence indicating that complement systemis activated by AP in CTX-injured muscle where it promotes subsequent muscle regeneration through macrophage recruitment and infiltration via C3a–C3aR signaling. During the aging process and after injury, efficient muscle regeneration is critical to maintain the mass and movement ability of skeletal muscles and alleviate muscle atrophy, and better understanding of the mechanisms and regulatory factors of muscle regeneration is important for human health. Our results in this study provide new insights into the mechanisms regulating muscle regeneration by identifying complement activation as an early priming event with significant influence on subsequent local inflammation and muscle regeneration. This finding offers a new perspective in the search for possible therapeutic interventions to enhance muscle regeneration.

## Methods

**Mice.** Wild-type C57BL/6J mice (WT) were purchased from National Animal Model Research Center (Nanjing, China), and mice carrying a null mutation for C3 (*C3*−/−), C4 (*C4*−/−), factor B (*Cfb*−/−), C3aR (*C3aR*−/−), or C5aR (*C5aR*−/−) in a C57BL/6J background were generated as previously described[37–39]. Mice were bred and kept under specific pathogen-free conditions at the animal facilities of Beijing Anzhen Hospital affiliated to the Capital Medical University. 8–10 week-old knockout mice and WT mice were used. All animal experimental protocols were approved by the Animal Subjects Committee of Capital Medical University.

**Muscle regeneration model.** Two kinds of injury-induced muscle regeneration model were used as described[32,40]. The tibialis anterior (TA) and the gastrocnemius muscles of anesthetized mice (100 mg kg−1, 1% pentobarbital sodium, i.p.) were injected with 30 μl and 60 μl of 10 μM cardiotoxin (Sigma, St. Louis, MO) or 50% glycerol solution (Sigma, St. Louis, MO) respectively. At different time points after injury, three to four mice were sacrificed by cervical dislocation under anesthesia and muscles were harvested. TA muscles were mounted in OCT and frozen in isopentane chilled with liquid nitrogen and stored at −80 °C. Gastrocnemius muscles were frozen in liquid nitrogen for mRNA and protein extraction.

**Serum reconstitution and measurement of AP activation.** Serum reconstitution was carried out as described previously[25]. Sera were obtained from WT mice, then 200 μl sera per *Cfb*−/− mouse were injected intravenously. Muscles were injured by CTX after serum reconstitution. To examine the efficiency of serum reconstitution[41], sera were obtained at 0, 1, 2, 4, 8 h after serum reconstitution, diluted with

Mg$^{2+}$-ethyleneglycol tetraacetic acid (EGTA) for 1:50. Then the enzyme-linked immune sorbent assay of complement activation was performed as previous described using 96 well plates coated with LPS (2 μg per well) at 4 °C overnight. After blocking with bovine serum albumin (BSA) (10 mg ml$^{-1}$) for 1 h, plates were washed with phosphate-buffered saline (PBS) for three times. Diluted mouse serum (50 μl per well) was incubated on plates at 37 °C for 1 h followed by detection of plate-bound activated C3 using HRP anti-mouse C3 antibody (1:4,000). Then TMB substrate Kit (Cell Signaling Technology) was used to detect activated C3. LPS-coated wells without serum incubation were used as background controls.

**Generation of bone marrow chimeric mice**. Eight-week-old WT and C3aR−/− male recipient mice were pre-conditioned for 2 weeks with pH = 2.0 drinking water containing 100 mg L$^{-1}$ levofloxacin and 100 mg L$^{-1}$ fluconazole. Bone marrow cells were collected from femurs and tibias of WT or C3aR−/− mice and re-suspended in RPMI-1640 with 2% FBS and 5 U ml$^{-1}$ heparin at a density of 1 × 10$^8$ cells per ml and kept on ice. 4 h after irradiation with 10 Gy X-rays, recipient mice were intravenously injected with 1 × 10$^7$ bone marrow cells, and then the mice were kept in a specific pathogen-free environment for 8 weeks with pH=2.0 drinking water containing 100 mg L$^{-1}$ levofloxacin and 100 mg L$^{-1}$ fluconazole to reconstitute their bone marrow. Four groups of chimeric mice were generated: WT to WT, C3aR−/− to WT, WT to C3aR−/− and C3aR−/− to C3aR−/−. To evaluate efficiency of bone marrow reconstitution, bone marrow cells were collected from femurs of representative mice at the time of sacrifice, and genomic DNA was isolated and used for genotyping.

**Histological and immunohistochemical analysis**. Serial, transverse cryosections (7 μm thick) of the midbelly region of frozen TA muscles were cut at −20 °C using a CM1950 Frigocut (Leica, Wetzlar, Germany), and all cryosections were kept at −80 °C until further studies. Picrosirius red staining was used to detect the collagen deposition. To analyze the cross-sectional area (CSA) of myofibers that had been initially immunostained for FITC conjugated wheat germ agglutinin (1:50 diluted, Sigma, Santa Cruz, CA). Other antibodies used for immunohistochemical staining and FACS, are listed in Supplementary Table 1. The uninjured muscles (day 0) were used as negative controls. The muscle tissue sections were incubated with primary antibodies at 4 °C overnight after blocked with serum, then secondary antibodies incubated at 37 °C for 30 min and detected with 3,3′-diaminobenzidine (DAB). Images obtained from each muscle section (ECLIPSE 90i, Nikon, Japan), and analyzed by NIS-Elements Br 3.0 software.

**Flow cytometry analysis**. Gastrocnemius muscles were dissociated in PBS containing collegenase I (200 U ml$^{-1}$) and dispase II (2.4 U ml$^{-1}$) at 37 °C for 30 min, The single-cell suspension was filtered, centrifuged, and resuspended in PBS. Cells were blocked with CD16/32 antibody and then incubated with antibodies diluted in PBS for 30 min at 4 °C. All the antibodies used in flow cytometry are listed in Supplementary Table 1. By anti-CD45 staining, cells were gated to CD45-positive bone marrow-derived cells and CD45-negative muscle resident cells. Then by anti-CD11b, anti-CD3, and anti-B220 staining, cells in CD45-positive gate was divided into CD11b-positive monocytes cells, CD3-positive T cells and B220-positive B cells. The cells in CD11b-positive gate were further divided into Gr1$^+$F4/80$^-$neu-trophils, Gr1$^{hi}$F4/80$^+$ pro-inflammatory macrophages and Gr1$^{low}$F4/80$^+$ anti-inflammatory macrophages[8]. By anti-CD31, anti-Sca-1, and anti-α7-integrin staining, cells in CD45-negative cells were divided into CD31$^+$ endothelia cells, CD31$^-$Sca-1$^+$ fibro/adipogenic progenitors, and CD31$^-$α7-integrin$^+$ myoblast. Expression of surface molecules was analyzed by flow cytometry (BD LSRFortessa) and associated software (BD FACSDiva Software).

**Cell sorting**. For sorting of CD45$^+$CD3$^+$ T cells, CD45$^+$CD11b$^+$ monocytes, and CD45$^-$ muscle resident cells in injured muscle, tissues were minced and digested to single-cell suspension, then cells were labeled and sorted by Beckman Coulter MoFloTM XDP. In some experiments for the sorting of CD11b$^+$ cells, CD11b microbeads, LS column and MidiMACS Separators(Miltenyi Biotec., Auburn, CA) was used according to the instruction manual. The purity of sorted cell populations was verified by flow cytometry.

**RNA-seq analysis**. RNA-Seq analysis was carried out using RNA samples converted into individual cDNA libraries using Illumina TruSeq methods employing single reads of 50 base-lengths sequenced at 20–30 million read depths using the Illumina HiSeq 2000 instrument. Differential and significant gene expression analysis was carried out using gene-level RPKM (Reads Per Kilobase of exon model per Million mapped reads) expression levels. Genes were selected using the criteria of an absolute expression level greater than 1 RPKM in either WT or C3aR−/− samples with at least 1.5-fold higher expression in WT or C3aR−/−. Gene lists relative enrichments for various functional associations was determined using KEGG database. Gene interaction network was analyzed by STRING software.

**Proliferation assay**. In vivo, to detect proliferating cells at day 3 and day 5 after injury, mice were injected with 5-bromo-2′-deoxyuridine (BrdU, Sigma; 100 mg kg$^{-1}$) 18 h before harvest. BrdU inmmunostaining was performed with primary antibody of mouse monoclonal BrdU (1:400, Zhongshan Jinqiao, Beijing, China) at 4 °C overnight after nucleus denatured by 2 M HCl and 0.1 M sodium tetraborate (pH 8.5), and then secondary antibody was incubated at 37 °C for 30 min for DAB detection or goat anti-mouse Alexa Fluor 488 antibody (Invitrogen) was used for immunofluorescence staining. To detect the proliferation of macrophages, APC BrdU flow kit (BD)was used according to instruction manual.

**TUNEL staining**. The TUNEL procedure was performed using the In Situ Apoptosis Kit (Promega, Madison, MI). After fixation in 4% paraformaldehyde for 5 min, the slides were incubated in equilibration buffer for 10 min before application of the rTdT reaction mix to the tissue sections on each slide, which were then incubated at 37 °C for 60 min, followed by 2× SSC washes three times. The slides were stained with DAPI and coverslipped.

**mRNA extraction and quantitative real-time PCR**. Total mRNA was extracted from gastrocnemius muscles or cells using TRIzol (Invitrogen, Carlsbad, CA) as described previously[38]. The concentration of mRNA was measured by Nanodrop ND-2000C (Thermo Scientific), then 2 μg mRNAs were reversely transcribed using the reverse transcription kit (Promega, Madison, WI). The mRNA levels of genes were analyzed by qRT-PCR, which were performed with 2× SYBR master mix (Takara, Otsu, Shiga), using BIO-RAD CFX CONNECT system (Bio-Rad). DNA primer sequences are detailed in Supplementary Table 2. Relative expression levels of genes were calculated from cycle threshold values using β-Actin as an internal control. (Ct; gene relative expression = 2$^{[Ct\ (\beta\text{-Actin})\text{-Ct (target gene)}]}$.

**Statistical analysis**. Results are expressed as mean ± s.e.m. unless stated otherwise. Statistical comparisons between two groups were evaluated by unpaired Student's t-test, two-tailed. For all statistical tests, a probability (P) value <0.05 was considered to indicate statistical significance.

**Data availability**. Sequence data that support the findings of this study have been deposited in Array Express with the primary accession code E-MTAB-5886. The other data that support the findings of this study are available from the corresponding author upon reasonable request.

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

## Acknowledgements

This research is supported by grants from the National Natural Science Foundation of China (81430050, 81230006 to J.D., 81500265 to C.Z.) and the National Institutes of Health (AI44970 and AI085596 to W.-C.S.).

## Author contributions

C.Z. designed the experiments, conducted the experiments, analyzed the data and wrote the manuscript. J.D. and W.-C.S. designed the experiments and edited the manuscript. C.W., Y.L., T.M., C.L., W.C. conducted experiments and analyzed data.

## Additional information

**Competing interests:** The authors declare no competing financial interests.

