## [Peer Review file · Nature Communications]

Reviewers' comments:

Reviewer #1 (Remarks to the Author):

The study by Zhang et al. investigates the role of complement alternative pathway and C3a/C3aR and CCL5 axis in macrophage recruitment during skeletal muscle regeneration. Apart minor points (listed below), the study is well conducted and the results are convincing. The data show that C3a/C3aR axis is required for skeletal muscle regeneration, and that this defect is rescued by early CCL5 i.m. injection. This signaling axis is completely novel in tissue injury and muscle regeneration.

Main issue:

1) However, to reach the level of Nat Commun standard, some mechanistic insights should be provided. The main concern is the lack of evidence of the link between C3a/C3aR and CCL5 in the process of monocyte recruitment. CCL5 has already been involved in muscle regeneration (Kohno et al. Muscle and Nerve 2011), and the reduction of half of macrophages has already been shown to delay muscle regeneration (Chazaud, Munoz-Canoves, Shireman, Zhou groups). Thus while the novelty of the C3a/C3aR/CCL5 axis seems very exciting, and appears to play a role in muscle regeneration, its specific involvement on monocyte attraction is not demonstrated here.

Important issues:

2) Magnetic beads at day 1 in the CTX model using CD11b allow the isolation of both macrophages and neutrophils. This has to be stated that the transcriptome analysis includes expression of genes in neutrophils.

3) The absolute number of monocytes/macrophages at day 1 should be calculated in order to confirm the decrease of infiltration (e.g. Figure 3F 3G) and not by a percentage of isolated cells. Indeed, the total number of isolated cells can vary from one muscle to the other, depending on the procedure and the gating strategy.

4) The number of neutrophils and of other leukocytes (CD45+) cells should be assessed in the various conditions to validate the specificity of the C3aR/CCL5 on macrophage recruitment (and not alteration of other leukocytes).

5) The direct role of C3a/C3aR and CCL5 axis in the recruitment of macrophages should be assessed at least in vitro, on the recruitment itself (e.g. by using chemotactic chambers with muscle extract, or other devices).

6) From day 2 macrophages acquire a Ly6Cneg phenotype and start to proliferate. While the authors investigated the absence of effect of C3a on macrophage apoptosis, they should also check that C3a or CCL5 do not alter macrophage proliferation. This will allow to confirm the specificity of C3aR/CCL5 axis on the recruitment of the macrophages and not on their later expansion. Indeed, the decreased in the number macrophages (in FBKO animals) is much more pronounced for Ly6Cneg than for Ly6Cpos cells (Fig3H).

Minor points:

7) Introduction: authors state "However, it is not clear what signal initiates the influx of monocytes to injured muscle tissues." It would be fair to introduce here the big work that has been done on MCP1 in muscle regeneration which shows that without MCP1, the muscle is hardly regenerating because of the lack of monocyte/macrophage recruitment, the phenotype being much more important than that observed in the present study in FB KO or C3aR KO animals. This point should be discussed anyway in the discussion.

8) While introducing the classical, lectin and alternative pathways in the result section, these pathways and their main molecular determinants should be briefly introduced (or in the introduction section) for readers who are non-specialists in complement biology.

9) Citations of references have to be edited (two series of brackets, space lacking, etc.).

10) Spelling has to be checked (myogen instead of myogenin, page 6), some sentences have to be rephrased (e.g. To confirm the role of C5a/C5aR signaling lacking in muscle regeneration... page 8).

11) The order of supplemental Figures is not logical. Figure S5 is cited before S3 and S4, leading to confusion.

Reviewer #2 (Remarks to the Author):

The manuscript titled "complement C3a signalling facilitates skeletal muscle regeneration by regulating monocyte function and trafficking" describes the role of complement signaling through the alternative pathway in promoting skeletal muscle regeneration by regulating monocyte function and trafficking. Early muscle regeneration in C3^{-/-} mice was impaired demonstrating that complement pathway is indeed involved in stimulating muscle regeneration following injury. Further experiments with Factor B^{-/-} and C4^{-/-} animals revealed that only the alternative complement cascade appeared to be critical for regeneration. Consistent with these results there were fewer infiltrating CD45 and F4/80 cells in the regenerating tissue of Factor B^{-/-} animals. The author's also demonstrated that C3aR^{-/-} animals demonstrated a similar phenotype to the Factor B^{-/-} and C3^{-/-} during regeneration. In addition to smaller fibre CSA in these knockout animals there were also fewer proliferating myoblasts in the early phases following injury in the C3aR^{-/-} animals. Finally, CCL5 administration rescued deficiencies in early muscle regeneration and inflammatory cell infiltration in C3aR^{-/-} animals. Collectively, this manuscript includes a number of elegantly executed experiments that quite thoroughly investigate the issue of complement pathway involvement in muscle regeneration. I do, however have some concerns that should be addressed before this manuscript should be considered further.

Comments:

1. This is an excellent manuscript that has thoroughly investigate the issue of complement pathway involvement in skeletal muscle regeneration. The major shortcoming of this manuscript is that all regeneration experiments appear to have been terminated at 15d following injury (although there is reference to a 30d time-point on page 8 but there is no data to support the statement. The difficulty with this is that it remains unknown whether the alternative complement pathway is actually critical for regeneration since the experiments never followed the repair process through to complete repair. It may indeed be that complement pathway plays some role in recruiting monocytes into injured muscle and in the absence of alternative complement signaling there is a "delay" in regeneration. Without a full characterization of regeneration it is difficult to say that complement signaling is "critical" for muscle regeneration. This limitation should be considered and the discussion should be revised accordingly – for example, the final paragraph of the discussion is a significant stretch given this particular limitation. Having said this, the experiments clearly and convincingly demonstrate that early regeneration is impacted by the loss of alternative complement signaling.

2. MyoD and myogenin mRNA expression were assessed in the FB^{-/-} animals and were shown to be lower. Is the data available for the C4^{-/-} animals? This data should be referenced to compliment the CSA data. Also was neonatal or embryonic MHC mRNA or protein expression measured during regeneration? Genes associated with fibrosis were highlighted (α SMA and TGF- β 1) and shown to be increased in FB^{-/-} animals – was MHC expression lower in these animals?

Minor comments:

Line 178 – should read "To determine if"

Line 250 – should read "Most" not "Mostly"

Line 270 – should read "myofibers" not "myofiber"

Line 292 – space between "expressed" and "a" needed and should read "damaged" not "damage"

Line 301 – should read "myoblasts" not "myoblast"

Reviewer #3 (Remarks to the Author):

The paper by Zhang et al. aims to elucidate functions of the complement system in regeneration of skeletal muscles in a model of cardiotoxin injury. The investigators have demonstrated that various complement deficiencies are associated with the reduced myofiber cross-section area (CSA) at day 15 after toxin injection, which was interpreted as indicative for impaired regeneration. The reduced CSA in factor B deficient mice has been attributed to the participation of the alternative complement pathway in this process. The phenotypic alterations have been correlated with the reduced production of chemokines and reduced recruitment of macrophages to regenerating tissue. The reconstitution of C3aR^{-/-} with CCL5 has restored regeneration pointing to a crucial role of this chemokine in the complement-mediated mechanisms partaking in regeneration. Although these observations have potential to be significant and innovative some of experimental approaches and interpretation of data raise questions and concerns.

Major issues:

1. It remains unclear what is an impact of complement on the extent of injury caused by cardiotoxin. It is possible that the observed differences between complement-deficient and wild-type mice in the tested surrogates of regeneration result from different extent of tissue injury caused by injection of cardiotoxin. Is this muscle injury reduced in complement deficient mice, and therefore, these muscles regenerate less in complement deficient mice? The extent of injury is known to have impact on a pace of regenerative/repair response. This is a legitimate question given the contribution of the complement to tissue injury in several disease models, especially myocardium. The extent of fibrosis may also reflect the extent of the initial injury.
2. With a few exceptions, the evaluation of muscle regeneration for most of key experiments rely on a single readout (CSA) in a single time point. This analysis does not consider complexity of the process, which involves time-dependent changes in: satellite cell proliferation, morphology of muscle fibers (size and number), quality and quantity of inflammatory infiltrate, and remodeling of vasculature and fibrosis. Time course experiments that consider these variables in key complement-deficient strains would strengthen this work.
3. Reproducing the observed phenotype in a different injury model in C3aR^{-/-} will validate the obtained results.
4. The experiments with bone marrow transplantation are not informative as C3aR-deficiency is associated with defective engraftment of hematopoietic stem progenitor cells (Leukemia. 2009 Aug;23(8):1455-61. doi: 10.1038/leu.2009.73. Epub 2009 Apr 9.)
5. What are CD45 negative cells that can be retrieved into single cell suspension from regenerating muscles? It is conceivable that the most cells that can easily get into single cell suspension, even after the enzymatic digestion, will be infiltrating inflammatory cells (CD45+). However, the percentages reported in the manuscript are low. Showing gating strategy and the representative FACS plots will help.
6. Can the authors provide reference/s for pro- vs anti-inflammatory macrophages that are classified based on Gr-1 expression?
7. Based on the figure description it appears that most of experiments were performed once with modest number of mice per cohort.
8. More thorough discussion of roles of complement in regeneration could be provided in the introduction, key seminal papers are not cited. Good summary of functions of complement in regeneration can be found in Semin Immunol. 2013 Feb; 25(1): 29–38.
9. Finally, no cellular/molecular mechanisms that can potentially link complement, macrophages and regeneration are explored.

Minor points:

1. The figure descriptions lack details regarding statistical analysis used and p values.
2. One way ANOVA is typically used for comparing more than two means. For comparing two means the authors may consider alternative approaches
3. Details regarding software used for FACS and gating strategies are missing.

Reviewers' comments:

Reviewer #1 (Remarks to the Author):

The study by Zhang et al. investigates the role of complement alternative pathway and C3a/C3aR and CCL5 axis in macrophage recruitment during skeletal muscle regeneration. Apart minor points (listed below), the study is well conducted and the results are convincing. The data show that C3a/C3aR axis is required for skeletal muscle regeneration, and that this defect is rescued by early CCL5 i.m. injection. This signaling axis is completely novel in tissue injury and muscle regeneration.

We thank the reviewer for these positive comments.

Main issue:

1) However, to reach the level of Nat Commun standard, some mechanistic insights should be provided. The main concern is **the lack of evidence of the link between C3a/C3aR and CCL5 in the process of monocyte recruitment**. CCL5 has already been involved in muscle regeneration (Kohno et al. Muscle and Nerve 2011), and the reduction of half of macrophages has already been shown to delay muscle regeneration (Chazaud, Munoz-Canoves, Shireman, Zhou groups). Thus while the novelty of the C3a/C3aR/CCL5 axis seems very exciting, and appears to play a role in muscle regeneration, its specific involvement on monocyte attraction is not demonstrated here.

Thank you for this suggestion. Our previous data shown that recombinant C3a or C3aR agonist stimulation could increase the expression of CCL5 in bone marrow derived monocytes (Supplemental Figure 7A and 7B). To add the evidence of the link between C3a/C3aR and CCL5 in the process of monocyte recruitment, we performed the transwell co-culture experiments. WT or C3aR^{-/-} bone marrow derived monocytes (5×10^5 /well) were seeded in the lower chamber and WT monocytes (5×10^4 /well) were seeded in the upper chamber. Then WT or C3aR^{-/-} monocytes were stimulated by C3aR agonist (1 μ M) for 12 hours and migrated cells were counted by DAPI staining (blue). Data shown that C3aR activated WT monocytes could recruit more monocyte from upper chamber than C3aR^{-/-} monocytes. After CCL5 was neutralized by anti-CCL5 antibody (10 μ g/ml), the C3aR activated WT monocytes recruited less monocytes.

These data were added in the new supplemental Figure 7D-7E.

D

Important issues:

2) Magnetic beads at day 1 in the CTX model using CD11b allow the isolation of both macrophages and neutrophils. This has to be **stated that the transcriptome analysis includes expression of genes in neutrophils.**

Thank you for this suggestion. We performed FACS analysis for CD11b cells at day 1 in the CTX model, data shown that in the CD11b⁺ cells, ~30% cells were Gr1⁺F4/80⁻ neutrophils and ~70% were Gr1⁺F4/80⁺ macrophages. Anti-CD11b magnetic beads were used to sort infiltrated macrophages and neutrophils from WT and C3aR^{-/-} mouse muscles at day 1 after injury, and profiled mRNA expression by RNA-seq and bioinformatics approach. This sentence was added at the page 9 paragraph 3 of the revised manuscript.

3) The **absolute number of monocytes/macrophages at day 1 should be calculated** in order to confirm the decrease of infiltration (e.g. Figure 3F 3G) and not by a percentage of isolated cells. Indeed, the total number of isolated cells can vary from one muscle to the other, depending on the procedure and the gating strategy.

As suggested by the reviewer, we used the counting beads (eBioscience, cat. 01-1234-42) to calculate the number of monocytes/macrophages in WT and Factor B^{-/-} muscle at day 1 and day 3 after injury. And these data was added in new Figure 3G. We also counting the numbers of macrophages in WT, C3aR^{-/-} and C5aR^{-/-} muscle at 3 day after injury, and these data were added in new Figure 4A.

4) **The number of neutrophils and of other leukocytes (CD45⁺) cells** should be assessed in the various conditions to validate the specificity of the C3aR/CCL5 on macrophage recruitment (and not alteration of other leukocytes).

As suggested by the reviewer, we also calculated the number of total CD45⁺ leukocytes, Gr1⁺F4/80⁻ neutrophils, CD3⁺T cells and B220⁺B cells in WT and Factor B^{-/-} muscle at day 1 and day 3 after CTX injury. Data shown that there was no difference in the number of Gr1⁺F4/80⁻ neutrophils, CD3⁺ T cells and B220⁺ B cells between WT and Factor B^{-/-} muscle. The number of in Factor B^{-/-} muscle was also less than WT muscle. These data were added in new figure 3G and 4A.

Figure 3:

Figure 4:

5) The direct role of C3a/C3aR and CCL5 axis in the recruitment of macrophages should be assessed at least in vitro, on the recruitment itself (e.g. **by using chemotactic chambers** with muscle extract, or other devices).

As suggested by the reviewer, we assessed the recruitment of monocytes by using chemotactic chambers with WT and C3aR^{-/-} muscle extract at day 1 after CTX injury. Data shown that C3aR^{-/-} muscle extract recruited less monocytes than WT muscle, the CCL5 concentration of C3aR^{-/-} muscle extract culture supernatant was also less than that of WT muscle. This data was added in the new Figure 6B and supplemental Figure 7F.

Figure 6

supplemental Figure 7

6) From day 2 macrophages acquire a Ly6C^{neg} phenotype and start to proliferate. While the authors investigated the absence of effect of C3a on macrophage apoptosis, they should also check

that C3a or CCL5 do not alter macrophage proliferation. This will allow to **confirm the specificity of C3aR/CCL5 axis on the recruitment of the macrophages and not on their later expansion**. Indeed, the decreased in the number macrophages (in FBKO animals) is much more pronounced for Ly6Cneg than for Ly6Cpos cells (Fig3H).

The reviewer's point is well taken. BrdU flow staining was used to assess the proliferation of macrophages at day 3 after CTX injury (BrdU was injected i.p. at 2 day after injury). As shown in new supplemental Figure 3D, in WT and C3aR^{-/-} muscle, after gated in CD11b⁺F4/80⁺ macrophages, there was no difference in the ratio of S phase macrophages and of G2/M phase macrophages.

supplemental Figure 3

Minor points:

7) Introduction: authors state "However, it is not clear what signal initiates the influx of monocytes to injured muscle tissues." **It would be fair to introduce here the big work that has been done on MCP1 in muscle regeneration** which shows that without MCP1, the muscle is hardly regenerating because of the lack of monocyte/macrophage recruitment, the phenotype being much more important than that observed in the present study in FB KO or C3aR KO animals. This point should be discussed anyway in the discussion.

We agree with the reviewer, indeed, several chemokines, such as MCP-1/CCL2^{1, 2}, MIP-1 α /CCL3³, CXCL12⁴, CXCL16⁵, CX3CL1⁶, was reported to participate in muscle injury and regeneration, which could recruit the circulation monocytes to injured muscle and promote the migration of myoblast. But how the increased expression of these chemokines induced in monocyte and muscle cells was still unknown. Our previous study found that cytokine interleukine-6 could increase the expression of CCL2 and CCL3 in macrophages, which could promote muscle regeneration⁷. In this study we have found that complement system was activated immediately after muscle injury, and complement C3a could increase the expression CCL5 in monocytes, which could recruit more macrophages to injured muscle.

These sentences were added at page 12 in revised manuscript.

8) While introducing the classical, lectin and alternative pathways in the result section, **these pathways and their main molecular determinants should be briefly introduced (or in the introduction section)** for readers who are non-specialists in complement biology.

Three main pathways can activate the complement system: classical, lectin, and alternative⁸. The classical pathway is activated when C1q binds to antibody attached to antigen, activating C1r and C1s, which cleave C4 and C2. The lectin pathway is activated when mannose-binding lectin (MBL) activating the MBL-associated serine proteases (MASPs) and again cleaving C4 and C2. C4 and C2 cleavage products form the classical and lectin pathway C3 convertase, C4bC2a, which

cleaves C3 into C3b and C3a. C3b can associate with C4bC2a to form the C5 convertase of the classical and lectin pathways, C4bC2aC3b. The alternative pathway (AP) is activated when C3 undergoes spontaneous hydrolysis and forms the initial AP C3 convertase, C3(H₂O)Bb, in the presence of Factors B and D, leading to additional C3 cleavage and eventual formation of the AP C3 convertase C3bBb and AP C5 convertase (C3bBbC3b).

This information was added at the page 5 in the revised manuscript.

9) Citations of references have to be edited (two series of brackets, space lacking, etc.).

As the reviewer's suggestion, we reformed the citations of references.

10) Spelling has to be checked (myogen instead of myogenin, page 6), some sentences have to be rephrased (e.g. To confirm the role of C5a/C5aR signaling lacking in muscle regeneration... page 8).

Thanks for the suggestion. We have rephrased the sentences and Modified the wrong spelling.

11) The order of supplemental Figures is not logical. Figure S5 is cited before S3 and S4, leading to confusion.

We are sorry for this mistake, we reordered the supplemental figures, the Figure S5 was named new Figure S3, and the Figure S3 was new Figure S4, the Figure S4 was new Figure S5.

Reviewer #2 (Remarks to the Author):

The manuscript titled “complement C3a signalling facilitates skeletal muscle regeneration by regulating monocyte function and trafficking” describes the role of complement signaling through the alternative pathway in promoting skeletal muscle regeneration by regulating monocyte function and trafficking. Early muscle regeneration in C3^{-/-} mice was impaired demonstrating that complement pathway is indeed involved in stimulating muscle regeneration following injury. Further experiments with Factor B^{-/-} and C4^{-/-} animals revealed that only the alternative complement cascade appeared to be critical for regeneration. Consistent with these results there were fewer infiltrating CD45 and F4/80 cells in the regenerating tissue of Factor B^{-/-} animals. The author's also demonstrated that C3aR^{-/-} animals demonstrated a similar phenotype to the Factor B^{-/-} and C3^{-/-} during regeneration. In addition to smaller fibre CSA in these knockout animals there were also fewer proliferating myoblasts in the early phases following injury in the C3aR^{-/-} animals. Finally, CCL5 administration rescued deficiencies in early muscle regeneration and inflammatory cell infiltration in C3aR^{-/-} animals. Collectively, this manuscript includes a number of elegantly executed experiments that quite thoroughly investigate the issue of complement pathway involvement in muscle regeneration. I do, however have some concerns that should be addressed before this manuscript should be considered further.

We are encouraged by these positive comments.

Comments:

1. This is an excellent manuscript that has thoroughly investigate the issue of complement pathway involvement in skeletal muscle regeneration. **The major shortcoming of this**

manuscript is that all regeneration experiments appear to have been terminated at 15d following injury (although there is reference to a 30d time-point on page 8 but there is no data to support the statement. The difficulty with this is that it remains unknown whether the alternative complement pathway is actually critical for regeneration since the experiments never followed the repair process through to complete repair. It may indeed be that complement pathway plays some role in recruiting monocytes into injured muscle and in the absence of alternative complement signaling there is a “delay” in regeneration. **Without a full characterization of regeneration it is difficult to say that complement signaling is “critical” for muscle regeneration.** This limitation should be considered and the discussion should be revised accordingly – for example, the final paragraph of the discussion is a significant stretch given this particular limitation. Having said this, the experiments clearly and convincingly demonstrate that early regeneration is impacted by the loss of alternative complement signaling.

Thanks for this positive suggestion. We have performed the 30 day time-point to examine the muscle regeneration. As shown in the new Figure 3A-3C, the myofiber CSA of WT and C4^{-/-} mice could increase similarly to uninjured muscle, but the myofiber CSA of FB^{-/-} mice was still much smaller than uninjured muscle. As shown in the new Figure 4C-4E, the myofiber CSA of WT and C5aR^{-/-} mice could increase similarly to uninjured muscle, but the myofiber CSA of C3aR^{-/-} mice was still much smaller than uninjured muscle. These data demonstrated that the alternative pathway activation of complement system and C3a-C3aR pathway is critical for muscle regeneration.

Figure 3:

Figure 4:

2. MyoD and myogenin mRNA expression were assessed in the FB^{-/-} animals and were shown to be lower. **Is the data available for the C4^{-/-} animals?** This data should be referenced to

compliment the CSA data. **Also was neonatal or embryonic MHC mRNA or protein expression measured during regeneration?** Genes associated with fibrosis were highlighted (α SMA and TGF- β 1) and shown to be increased in FB-/- animals – was MHC expression lower in these animals?

The MyoD and myogenin mRNA expression were also assessed in the C4-/- animals, As shown in the new supplemental Figure 1A, there was no difference in WT and C4-/- muscle at day 5 after injury. And the embryonic MHC mRNA was assessed in WT, C4-/- and FB-/- muscle at 5 day after CTX injury, the embryonic MHC mRNA in FB-/- muscle, but not in C4-/- muscle, was lower than WT muscle (supplemental Figure 1B).

supplemental Figure 1:

Minor comments:

Line 178 – should read “To determine if”

Line 250 – should read “Most” not “Mostly”

Line 270 – should read “myofibers” not “myofiber”

Line 292 – space between “expressed” and “a” needed and should read “damaged” not “damage”

Line 301 – should read “myoblasts” not “myoblast”

We have rephrased these spelling mistakes respectively.

Reviewer #3 (Remarks to the Author):

The paper by Zhang et al. aims to elucidate functions of the complement system in regeneration of skeletal muscles in a model of cardiotoxin injury. The investigators have demonstrated that various complement deficiencies are associated with the reduced myofiber cross-section area (CSA) at day 15 after toxin injection, which was interpreted as indicative for impaired regeneration. The reduced CSA in factor B deficient mice has been attributed to the participation of the alternative complement pathway in this process. The phenotypic alterations have been correlated with the reduced production of chemokines and reduced recruitment of macrophages to regenerating tissue. The reconstitution of C3aR-/- with CCL5 has restored regeneration pointing to a crucial role of this chemokine in the complement-mediated mechanisms partaking in regeneration. Although these observations have potential to be significant and innovative some of experimental approaches and interpretation of data raise questions and concerns.

We thank the reviewer for these positive comments.

Major issues:

1. **It remains unclear what is an impact of complement on the extent of injury caused by cardiotoxin.** It is possible that the observed differences between complement-deficient and wild-type mice in the tested surrogates of regeneration result from different extent of tissue injury

caused by injection of cardiotoxin. Is this muscle injury reduced in complement deficient mice, and therefore, these muscles regenerate less in complement deficient mice? The extent of injury is known to have impact on a pace of regenerative/repair response. This is a legitimate question given the contribution of the complement to tissue injury in several disease models, especially myocardium. The extent of fibrosis may also reflect the extent of the initial injury.

The muscle injury by Cardiotoxin (CTX) from cobra (*Naja naja atra*) venom was mainly cause necrosis of myofiber after muscle injection. CTX could lower the threshold for $\text{Ca}^{(2+)}$ -induced Ca^{2+} release in heavy sarcoplasmic reticulum fractions, and increase the permeability of myofiber plasma membranes, which induced the myofiber rupture⁸. By 24 hours the affected cells appeared as empty 'bags' containing only remnants of myofibrils and swollen mitochondria⁹. As an established mouse injury model, 30 μl CTX (10 μM) injection could injury almost all the myofiber, this injury course was not affected by complement system.

The muscle fibrosis was a temporarily alternative replacement of impaired regeneration, and if the extent of fibrosis could reflect the extent of the initial injury, the muscle injury should more severely in Factor B^{-/-} and C3aR^{-/-} mice for the increased muscle fibrosis in Factor B^{-/-} and C3aR^{-/-} mice.

2. With a few exceptions, **the evaluation of muscle regeneration for most of key experiments relay on a single readout (CSA) in a single time point**. This analysis does not consider complexity of the process, which involves time-dependent changes in: satellite cell proliferation, morphology of muscle fibers (size and number), quality and quantity of inflammatory infiltrate, and remodeling of vasculature and fibrosis. Time course experiments that consider these variables in key complement-deficient strains would strengthen this work.

We have added the statistical data of CSA at day 5 and day 30 in WT, C4^{-/-}, FB^{-/-}, C3aR^{-/-} and C5aR^{-/-} mice, which was also recommended by reviewer 2#. Furthermore, the proliferation of satellite cell at day 3 and day 5 after injury, the number of inflammatory cells infiltration at day 1 and day 3, and the fibrosis and fibrosis associated genes was examined in WT, FB^{-/-} and C3aR^{-/-} mice. These data were added to new figure 3 and figure 4 and figure 6.

3. Reproducing the observed phenotype **in a different injury model** in C3aR^{-/-} will validate the obtained results.

As suggested by the reviewer, we have performed a glycerol induced muscle injury in WT and C3aR^{-/-} muscle, and examined the muscle regeneration and fibrosis at 15 day after injury. As the data shown below, the CSA of regenerated myofiber was much smaller in C3aR^{-/-} muscle than that of WT muscle, and the fibrosis area increased in C3aR^{-/-} muscle. This data was added in the supplemental Figure 5B-5D.

supplemental Figure 5

4. **The experiments with bone marrow transplantation are not informative** as C3aR-deficiency is associated with defective engraftment of hematopoietic stem progenitor cells (Leukemia. 2009 Aug;23(8):1455-61. doi: 10.1038/leu.2009.73. Epub 2009 Apr 9.)

In the referred article, less leukocytes and platelet counts in circulation at 1-3 weeks after bone marrow transplantation revealed delayed engraftment of C3aR^{-/-} HSPCs as compared to WT HSPCs. But this article also shown that at 4 weeks after transplantation, there was no difference in leukocytes and platelet counts of WT and C3aR^{-/-} recipient mice (see below). Our muscle regeneration model was performed after 8 weeks after transplantation, to make a fully bone marrow recovery. To confirm the efficiency of bone marrow reconstitution, bone marrow cells of four types bone marrow chimera mouse (WT to WT, C3aR^{-/-} to WT, WT to C3aR^{-/-}, C3aR^{-/-} to C3aR^{-/-}, n=4 in each group) was collected and the C3aR mRNA level was accessed by realtime-PCR. As shown in the below data, the C3aR mRNA was higher in WT and C3aR^{-/-} recipient mice with WT bone marrow than that with C3aR^{-/-} bone marrow. This data was added in Figure 5A.

Figure 1. HSPCs from C3aR^{-/-} mice show defective engraftment

Wysoczynski M, Reza R, Lee H, Wu W, Ratajczak J, Ratajczak MZ. Defective engraftment of C3aR^{-/-} hematopoietic stem progenitor cells shows a novel role of the C3a-C3aR axis in bone marrow homing. *Leukemia*. 2009;23(8):1455-61.

Figure 5

5. What are CD45 negative cells that can be retrieved into single cell suspension from regenerating muscles? It is conceivable that the most cells that can easily get into single cell suspension, even after the enzymatic digestion, will be infiltrating inflammatory cells (CD45⁺). However, the percentages reported in the manuscript are low. **Showing gating strategy and the representative FACS plots will help.**

By FACS analysis shown below, at CTX 1 day after injury, the CD45 negative cells (~79% in all cells) were CD45⁻CD31⁺ endothelial cells (~2% in CD45⁻ cells), CD45⁻CD31⁻Sca-1⁺ fibro/adipogenic progenitors (~14% in CD45⁻ cells), CD45⁻CD31⁻Sca-1⁻α7-integrin⁺ myoblast (~10% in CD45⁻ cells) and other myofiber debris (~70% in CD45⁻ cells). This data was added in the new supplemental Figure 2E.

supplemental Figure 2

6. Can the authors **provide reference/s for pro- vs anti-inflammatory macrophages** that are classified based on Gr-1 expression?

Yes, we have added the reference “Mounier, R. et al. AMPK α 1 regulates macrophage skewing at the time of resolution of inflammation during skeletal muscle regeneration. *Cell metabolism* 18, 251-264 (2013).” as reference 8 at page 7 in the revised manuscript.

7. Based on the figure description it appears that **most of experiments were performed once with modest number of mice per cohort.**

The data in the figure represented one of two or three independent experiments.

8. **More thorough discussion of roles of complement in regeneration could be provided in the introduction,** key seminal papers are not cited. Good summary of functions of complement in regeneration can be found in *Semin Immunol.* 2013 Feb; 25(1): 29–38.

The role of complement proteins in tissue and organ regeneration has been reported in several literatures. For example, both C3a and C5a signaling pathways have been found to prime and facilitate liver regeneration after acute carbon tetrachloride injury in mice. C3a receptor signaling has been reported to be involved in lens regeneration in the newt. Complement C1q activates canonical Wnt signaling and promotes aging-associated decline in muscle regeneration independent with the classical complement activation. Alternative pathway of complement activation is required for efficient osteoclast differentiation and regeneration, and complement activation production C3a and C5a could regulate osteoclast differentiation by modulating local IL-6 production.

These sentences were added to page 3 paragraph 2 in the introduction section.

9. Finally, **no cellular/molecular mechanisms that can potentially link complement, macrophages and regeneration are explored.**

1. C3a receptor C3aR is a G protein coupled receptor. C3aR activation could increase the phosphorylation of AKT and NF- κ B as downstream signaling pathway, which was reported to promote the transcription of chemokine CCL5. This consequence was also supported by our data in supplemental Figure 7.

2. Chemokine CCL5 could recruit the macrophages into injured muscle, the decreased CCL5 in C3aR $^{-/-}$ muscle recruit less macrophages, which was evidenced by the transwell co-culture experiment. The data was added in Figure 6 and supplemental Figure 7.

3. The infiltrated macrophages could secrete other chemokines to recruited more macrophages, which amplify the inflammatory response in injured muscle. The infiltrated macrophages was reported to facilitate muscle regeneration by clearing the injured muscle by phagocytosis and producing growth factors and cytokines to promote myoblast proliferation and differentiation.

Minor points:

1. The figure descriptions **lack details regarding statistical analysis used and p values.**

The details information regarding statistical analysis used and p values was added to page 17 in the Methods section and figure description respectively.

Results are expressed as mean \pm s.e.m. unless stated otherwise. Statistical comparisons between two groups were evaluated by unpaired Student's t-test, two-tailed. A probability (P) value <0.05 was considered to indicate statistical significance.

2. One way ANOVA is typically used for comparing more than two means. For comparing two means the authors may consider alternative approaches

The unpaired two-tailed Student's t test was used to compare two means.

3. **Details regarding software used for FASC and gating strategies** are missing.

By anti-CD45 staining, cells were gated to CD45 positive bone marrow derived cells and CD45 negative muscle resident cells. Then by anti-CD11b, anti-CD3 and anti-B220 staining, cells in CD45 positive gate was divided into CD11b⁺ monocytes cells, CD3⁺ T cells and B220⁺ B cells. The cells in CD11b positive gate were further divided into Gr1⁺F4/80⁻ neutrophils, Gr1^{hi}F4/80⁺ pro-inflammatory macrophages and Gr1^{low}F4/80⁺ anti-inflammatory macrophages. By anti-CD31, anti-Sca-1 and anti- α 7-integrin staining, cells in CD45 negative cells were divided into CD31⁺ endothelia cells, CD31⁻Sca-1⁺ fibro/adipogenic progenitors and CD31⁻ α 7-integrin⁺ myoblast. Expression of surface molecules was analyzed by flow cytometry (BD LSRFortessa) and associated software (BD FACSDiva Software).

This information was added to page 16 in the Methods section.

1. Shireman, P.K. et al. MCP-1 deficiency causes altered inflammation with impaired skeletal muscle regeneration. *Journal of leukocyte biology* 81, 775-785 (2007).
2. Martinez, C.O. et al. Regulation of skeletal muscle regeneration by CCR2-activating chemokines is directly related to macrophage recruitment. *American journal of physiology. Regulatory, integrative and comparative physiology* 299, R832-842 (2010).
3. Yahiaoui, L., Gvozdic, D., Danialou, G., Mack, M. & Petrof, B.J. CC family chemokines directly regulate myoblast responses to skeletal muscle injury. *The Journal of physiology* 586, 3991-4004 (2008).
4. Griffin, C.A., Apponi, L.H., Long, K.K. & Pavlath, G.K. Chemokine expression and control of muscle cell migration during myogenesis. *Journal of cell science* 123, 3052-3060 (2010).
5. Zhang, L. et al. Chemokine CXCL16 regulates neutrophil and macrophage infiltration into injured muscle, promoting muscle regeneration. *The American journal of pathology* 175, 2518-2527 (2009).
6. Stromberg, A. et al. CX3CL1--a macrophage chemoattractant induced by a single bout of exercise in human skeletal muscle. *American journal of physiology. Regulatory, integrative and comparative physiology* 310, R297-304 (2016).
7. Zhang, C. et al. Interleukin-6/signal transducer and activator of transcription 3 (STAT3) pathway is essential for macrophage infiltration and myoblast proliferation during muscle regeneration. *The Journal of biological chemistry* 288, 1489-1499 (2013).
8. Ownby, C.L., Fletcher, J.E. & Colberg, T.R. Cardiotoxin 1 from cobra (*Naja naja atra*) venom causes necrosis of skeletal muscle in vivo. *Toxicon* 31, 697-709 (1993).
9. Fletcher, J.E., Jiang, M.S., Gong, Q.H., Yudkowsky, M.L. & Wieland, S.J. Effects of a cardiotoxin from *Naja naja kaouthia* venom on skeletal muscle: involvement of calcium-induced calcium release, sodium ion currents and phospholipases A2 and C. *Toxicon* 29, 1489-500 (1991).

REVIEWERS' COMMENTS:

Reviewer #1 (Remarks to the Author):

The authors answered to all the issues I raised. Thank you very much.

Reviewer #2 (Remarks to the Author):

Thank you for thoroughly addressing my comments. The manuscript is significantly improved over the initial iteration.

Reviewer #3 (Remarks to the Author):

The prior comments of the reviewers have been thoroughly addressed in the revised manuscript.

REVIEWERS' COMMENTS:

Reviewer #1 (Remarks to the Author):

The authors answered to all the issues I raised. Thank you very much.

Thanks for your positive comments.

Reviewer #2 (Remarks to the Author):

Thank you for thoroughly addressing my comments. The manuscript is significantly improved over the initial iteration.

Thanks for your positive comments.

Reviewer #3 (Remarks to the Author):

The prior comments of the reviewers have been thoroughly addressed in the revised manuscript.

Thanks for your positive comments.